# THE NEGATIVE PRETRAINING EFFECT IN SEQUENTIAL DEEP LEARNING AND THREE WAYS TO FIX IT

## ABSTRACT

*Negative pretraining* is a prominent sequential learning effect of neural networks where a pretrained model obtains a worse generalization performance than a model that is trained from scratch when either are trained on a target task. We conceptualize the ingredients of this problem setting and examine the *negative pretraining effect* experimentally by providing three interventions to remove and fix it. First, acting on the learning process, altering the learning rate after pretraining can yield even better results than training directly on the target task. Second, on the learning task-level, we intervene by increasing the discretization of data distribution changes from start to target task instead of "jumping" to a target task. Finally at the model-level, resetting network biases to larger values likewise removes negative pretraining effects, albeit to a smaller degree. With these intervention experiments, we aim to provide new evidence to help understand the subtle influences that neural network training and pretraining can have on final generalization performance on a target task in the context of *negative pretraining*.

## 1 INTRODUCTION

Lifelong learning (Thrun & Mitchell, 1995; Silver et al., 2013) holds both the promise of benefitting from past experiences and the challenge of having to continually adapt to new problem settings. The characteristics of this intriguing sequential learning problem make it a balancing act between retaining knowledge and adapting to new experiences referred to as the *stability-plasticity dilemma* (Mermillod et al., 2013). Similar to a coach or teacher, we want to understand when a sequence of tasks helps or hinders further learning and what the underlying mechanism are that influence this process given that the cost of training very large models from scratch such as GPT-3 (Brown et al., 2020) has increased considerably (Sharir et al., 2020). This setting of training a model on a sequence of tasks to maximize its performance on a target task we coin the *sequential learning problem*.

The existing literature on sequential learning sheds some light on the characteristics of this problem setting. In some cases, following a sequence of learning tasks can lead to better results than simply training on the target task from scratch (Bengio et al., 2009). Achille and Soatto (Achille et al., 2017) and Wang et al. (2019) contrast this picture by highlighting that neural networks can suffer from pretraining on some tasks. When learning data is corrupted for a sufficiently long period, even switching back to clean data does not allow the model to recover its original performance (Maennel et al., 2020). Until compared, it may unforuntely be unclear that a model is corrupted.

*Contribution:* We formalize the basic ingredients of *sequential learning* as following a path on a learning manifold as depicted in Fig. 1 and provide three ways to remove the negative pretraining effect that can occur for certain task changes in neural network training as depicted in Fig. 2. We establish experimentally for a variety of task changes and datasets that this final performance can be highly dependent on the way the chosen model is trained, the sequence of training tasks, and how the model is perturbed during the training process. First, we demonstrate that increasing learning rates after pretraining can remove negative pretraining effects. Secondly, we show that continuously changing from source to target task instead of "jumping" can reap significant benefits in performance and does not incur a deficit penalty. Thirdly, we display that resetting only the biases for each task in the training process appears to decrease negative pretraining effects. Finally we speculate on plausible reasons such as qualitatively different training behavior during the "high learning rate phase" (Lewkowycz et al., 2020) that may explain the observed effects.

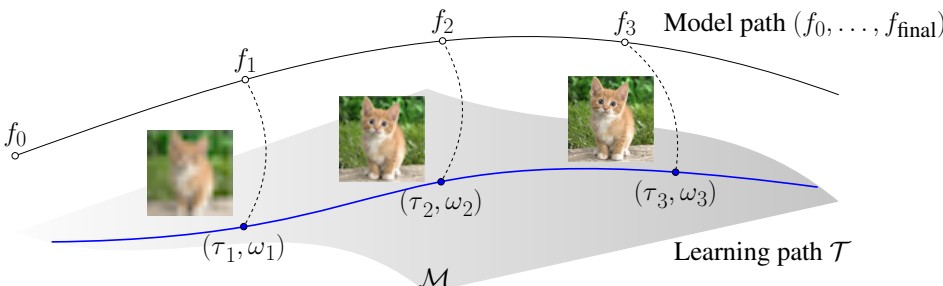

Figure 1: In the context of the *negative pretraining effect*, e.g. when pretraining on blurred images before training on unblurred images (Achille et al., 2017), we explore the effects of interventions on "learning paths", shown as the blue path on the learning manifold $\mathcal{M}$, on the *generalization performance* of neural networks. A point on the learning manifold is defined as a learning task $\tau := (p(x, y), \mathcal{L})$, composed of a data distribution of input and labels and a loss function, and a learning process $\omega$ which determines how a model $f$ is adapted given a task $\tau$. Tasks can vary by changing any part of the definition such as when the data distribution is changed along a *blurring* nuisance as depicted above with cat images. Learning processes can be adapted by changing their update equation, e.g. via learning rate changes $\Delta \eta$. The learning path $\mathcal{T}$ determines model path changes from initial to final trained model $f_{\text{final}}$. Via three interventions in the learning path we show that varying neural network learning paths can result in a removal of negative pretraining effects.

## 2 RELATED WORK

*Sequential learning* involves choosing a sequence of tasks and learning processes and following this learning path to improve performance on a single or set of target tasks. This strongly resembles the objectives of lifelong learning (Thrun & Mitchell, 1995), continual learning (Lesort et al., 2019) and meta-learning (Thrun & Pratt, 2012; Schaul & Schmidhuber, 2010; Hochreiter et al., 2001) and connects to many time-dependent learning aspects in relation to neural networks.

**Time-dependent learning:** We are aware of several time-dependent phenomena in training neural networks such as *catastrophic forgetting* (French, 1999), critical learning periods (Achille et al., 2017), and time sensitivity of regularization and data augmentation (Golatkar et al., 2019). Achille et al. (2017) show that by dropping out certain frequencies during the beginning of training, networks are unable to recover to full performance, even when those frequencies are reintroduced. Similarly, Liu et al. (2019) show that first optimizing on random labels ruins performance on clean data shown indefinitely to the same network afterwards. Recent information-theoretic analysis (Shwartz-Ziv & Tishby, 2017) suggests that neural network have distinct learning phases: First, a phase where network parameters grow in information content, before, in a second phase, self-regularizing and pruning away irrelevant information. More commonly known time-dependent or sequential heuristics are learning rate schedules (Darken & Moody, 1991), cyclical learning rates (Smith, 2017), and regularization annealing such as dropout annealing (Rennie et al., 2014), student-teacher model transfer (Vicente & Caticha, 1997), and pretraining (Erhan et al., 2009).

**Curriculum learning and generation:** Curriculum learning (Elman, 1993; Bengio et al., 2009) is the technique of training agents by showing them a progression of tasks from easier variants to harder examples to attempt to solve the final target task. Both the utility of transferring between similar or dissimilar tasks is debated, with some results pointing towards similarity (Achille et al., 2019; Ammar et al., 2014; Ritter et al., 2018) and others towards dissimilarity (Farquhar & Gal, 2018; Nguyen et al., 2019). Self-paced learning (Jiang et al., 2015) extends curriculum learning by letting the model itself influence the sequence of tasks (Graves et al., 2017). The order of tasks, the curriculum, is generally an *input* that is hand-designed by human experimenters and often based on heuristics to guide the scheduling (i.e. it may be easier to make an agent jump over small gaps before moving to larger ones (Heess et al., 2017)). Meta-learning (Schaul & Schmidhuber, 2010; Vilalta & Drissi, 2002) is similarly related by focusing on speeding up or enhancing performance on a target task by pretraining on other tasks to first *learn to learn*.

**Biological analogies:** It is through neuroplasticity (Dayan & Cohen, 2011) that animals and humans learn new skills and recover from changes or damage to their sensory organs or brain. Famously, Erismann and Kohler (Kohler & Pissarek, 1960) asked study participants to wear special mirror goggles which flipped the participants' visual field upside down as summarized in (Sachse et al., 2017). Remarkably, the participants visual perception adjusted after a few days by returning the visual field back to their "upright" standard perception. In contrast to this, there are other stimuli people learn to discard in childhood if they are not needed. Young children can differentiate phonemes of both their own and foreign languages; however, older children are better at distinguishing native phonemes and have more difficulty distinguishing foreign phonemes (Cheour et al., 1998; Kuhl et al., 1992). A similar effect of "perceptual narrowing" is found for 6-month-old and 9-month-old children when comparing the ability to discriminate between human faces to the ability to discriminate between primate faces (Pascalis et al., 2002). Younger children are again found to be less specialized to human faces than slightly older ones. Summarizing many results (Costandi, 2016), we currently know that neuroplasticity is "largest" at an early age yet generally does not end in adulthood. It is instead always present in reorganization processes involved in new experiences and when adjusting to injury.

## 3 SEQUENTIAL LEARNING SETTING

We are interested in better understanding what influences the outcome of a sequential learning problem. In the following, we say that a pretrained model has *plasticity* if it can generalize similarly well when trained on a target task as an uncorrupted model trained directly on the target task. If it cannot generlize similarly well, we refer to this as a *negative pretraining effect*. To facilitate further reading, we summarize essential elements such as the definition of a learning task $\tau$, constrained to supervised tasks in this work, the learning mapping $\omega$, the *learning path* as an ordered sequence of task-learning process pairs $\mathcal{T}$, and the setup of its repeated application in a *sequential learning process* $\Omega$ in Tab. 1 and visualize the overall setting in Fig. 1.

Table 1: Summary of notation, definitions, and terminology

| Symbol | Definition |
|---|---|
| $x \in \mathcal{X}, u \in \mathcal{U}, y \in \mathcal{Y}$ | Input, model output, and desired output |
| $p(x, y)$ | Joint probability distribution of input x and target or label y |
| $f : \mathcal{X} \times \Theta \to \mathcal{U}, \theta \in \Theta$ | Model function with parameters in parameter space |
| $\mathcal{L}(p, f) : (p, f) \mapsto \mathbb{R}_+$ | Loss objective of accumulated loss of model across distribution |
| $\mathcal{T} = ((\tau_1, \omega_1), ..., (\tau_n, \omega_n))$ | Learning path $\mathcal{T}$ as a sequence of tasks $\tau_i$ and learning processes $\omega_i$ |
| Task $\tau := (p, \mathcal{L})$ | **Definition 1** (Supervised task). *We define a supervised task as the optimization problem associated with the tuple of dataset distribution $p(x, y)$ and loss function $\mathcal{L}(p, f) : (p, f) \mapsto \mathbb{R}_+$, where in this work $\mathcal{L}(p, f) = \mathbb{E}_{p(x,y)}[\ell(f(x), y)]$ with individual data point loss $\ell : \mathcal{U} \times \mathcal{Y} \to \mathbb{R}_+$.* |
| Learn $\omega : (\tau, f) \mapsto f$ | **Definition 2** (Learning). *We define learning as a mapping of a task and an initial model $f_{init}$ to a new learned model $f_{learn}$: $\omega : (\tau, f_{init}) \mapsto f_{learn}$* |
| $\Omega : (\mathcal{T}, f) \mapsto f$ | **Definition 3** (Sequential learning process). *We define a sequential learning process $\Omega$ of a learning path consisting of a sequence of ordered task-learn process pairs $\mathcal{T} = ((\tau_1, \omega_1), \ldots, (\tau_n, \omega_n))$ as the sequential application of the task-learn process pairs to an initial model $f_{init}$. $f_\Omega = \omega_n(\tau_n, \omega_{n-1}(\tau_{n-1}(\ldots, \ldots, f_{init}) =: \Omega(\mathcal{T}, f_{init})$* |

A task $\tau$ consists of a probability distribution $p(x, y)$ of data points and desired outputs as well as a loss function $\mathcal{L}$ that maps an input-output behavior of a model to a cost score. The learning mapping $\omega$ uses the information in task $\tau$ and initial model $f_{init}$ to train a model optimized for the task $f_{learn}$ until convergence. Applying the learning mappings and tasks $(\tau_i, \omega_i) \in \mathcal{T}$ sequentially starting from an initial model $f_{init}$ is summarized via the symbol $\Omega$. Based on these elements, we ask what influences the final generalization outcome on a target task when following a learning path of task-learn process pairs $\mathcal{T}$. Specifically, we examine these question via three interventions aimed at distinct ingredients of the learning process as visualized in Fig. 2.

Figure 2: We explore the *negative pretraining effect* for neural networks, where, as a default, a model $f$ is first trained via learning process $\omega$ on a task $\tau_0$, e.g. with blurred images, and later trained on the same task with sharp images $\tau_{\text{final}}$ yet never regains the performance of only training on sharp images. We investigate three different interventions to this negative pretraining scenario: A) we vary the learning rate $\eta$ during training on blurred and sharp images; B) Instead of "jumping" from blurred to sharp images, we insert additional tasks with a finer discretization of blurring; C) we reset the biases of the network optionally before training on the first and/or second task.

## 3.1 GENERAL EXPERIMENTAL BENCHMARK SETUP

We will focus on stochastic gradient descent (SGD) learning applied to convolutional neural networks. As a stopping criterion, we consider the scenario where training has already converged, where the validation loss cannot be decreased further by changing the model $f$ via SGD. Empirical task transfer often deals with tasks that are separate (Zamir et al., 2018), and while fruitful empirically, it becomes difficult to understand conceptually. Instead we employ controlled customized task changes, allowing us to better gauge the effect of task changes. With the basic building blocks in place, we ask in which cases, even with an infinite number of data points, previous tasks can hinder learning and generalization on new tasks. We examine three intervention scenarios acting on various levels of abstraction as visualized in Fig. 2:

A) **Learning rate changes:** Applying changes to the learning process, we observe how employing different learning rates both on the first and second task can influence pretraining effects.

B) **Task discretization:** Intervening on the task-level, we consider the influence of inserting more tasks between first and last task by discretizing the changes from first to last task.

C) **Model biases resetting:** Finally, we act on the model and examine how final performance is affected when resetting biases before tasks.

We deal with four visual *supervised classification* datasets of which we use three in every experimental section below thereby significantly extending smaller scale experiments by Achille et al. (2017):

- **CIFAR-10** (Krizhevsky, 2009): A 32x32 full-color dataset with 10 classes.
- **FashionMNIST** (Xiao et al., 2017): A 28x28, black-and-white dataset with 10 classes.
- **MNIST** (LeCun et al., 1998): A 28x28, black-and-white dataset with 10 classes.
- **SVHN** (Sermanet et al., 2012): A 32x32 full-color dataset with 10 classes.

For all experiments if not mentioned otherwise, we use Pytorch (Paszke et al., 2019) and run a ResNet-18 (He et al., 2016) with a fixed initial learning rate of 10e-4 with momentum 0.9 and weight decay 1e-3. The networks are initialized using the standard Pytorch layer intialization discussed in Appendix 8.2. Each task is trained until convergence, for a maximum of 200 epochs per task, using early stopping on a validation set with patience 50. As baseline, the model was trained *only* on the target task - the standard uncorrupted dataset.

## 4 HIGHER LEARNING RATES PREVENT A LOSS OF PLASTICITY

The learning rate schedule is arguably one of the most important hyperparameters in neural network training as attested to by many deep learning practitioners (Li et al., 2019; Nakkiran, 2020; Smith, 2017). In several works (Seong et al., 2018; Zhu et al., 2018; Kleinberg et al., 2018; Lewkowycz et al., 2020), a higher learning rate has been linked to better generalization due to its increased probability to escape local minima during optimization. By escaping sharp minima, the network is deemed more likely to converge to a flat minimum (Hochreiter & Schmidhuber, 1995), i.e. a minimum with a

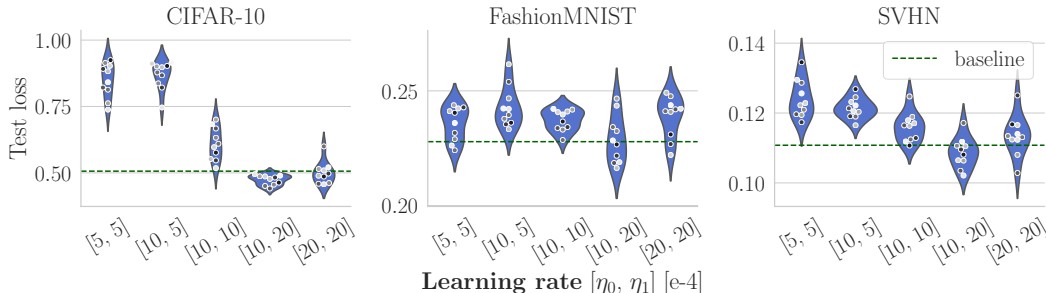

Figure 3: Final test loss on unblurred images as swarm and violin plot. The x-axis displays the learning rates on the first task with blurred images and the second task with unblurred images. The y-axis shows the test loss on the unblurred images with dots colored to distinguish different random seeds. The dotted line indicates the mean baseline performance across 3 seeds. A clear trend that negative pretraining can be overcome when increasing the learning rate is visible, particularly when moving from 10e-4 to 20e-4.

Hessian with lower maximal eigenvalue. Accordingly, we present a number of experiments detailing the effect of the learning rate on outcomes of neural network training after *negative pretraining*.

We keep the number of tasks, i.e. two, fixed and focus on the blurring transformation for all three datasets. For the first task, all images are downscaled to $1/4$ their size before rescaling to their original size. The second task has standard unblurred images. The default learning rate $\eta$ of 1e-3 is varied either for both tasks, meaning a constant learning rate over the learning path, or kept for the first and changed only for the second task. We picked 2e-3 as a higher and 5e-4 as a lower learning rate. Each experiment was performed for 10 random seeds. The baseline, trained directly on the standard uncorrupted data, is averaged across 3 random seeds. Results are shown in Fig. 3.

We observe the overall trend that higher learning rates appear to decrease the difference between models pretrained on blurred images and the baseline models trained on sharp images. Remarkably and consistently across datasets, training first with a learning rate 1e-3 and then 2e-3 is better in expectation than only training at the higher learning rate 2e-3. We conclude that in sequential learning, the precise order of learning rates across tasks can be crucial to obtain the best final learning loss.

## 5 SMOOTHLY CHANGING THE TASK CAN PREVENT A LOSS OF PLASTICITY

Reasoning that a more "continuous" change from source to target task provides the model with more opportunity to adapt, we provide an intervention that changes the *discretization of "jumping" from source to target task*. To create each learning path, we focus on effects of three common image transformations: *blurring*, *contrast*, and *random shifts*, as visually depicted in the appendix in Fig. 7. *Blurring* applies a resizing operation to each image, before resizing it back to the original size (as done in (Achille et al., 2017)). *Contrast* changes an image's channel attributes: A contrast of $0\%$ results in a fully gray image[1], whereas a contrast of $100\%$ results in the standard image. Finally, *random shift* changes the *maximum width* of the random affine transformation applied to the image: Each component of the shift is sampled from $[-a, a]$ and then applied to the image, where $a$ depends on the task parameterization. For reproducibility further details are provided in appendix 8.1.

**Discretization of paths:** Traditionally, curriculum learning is concerned only with the *sequence* of tasks. In contrast, to test plasticity, we are additionally concerned with how discretely we traverse the space from source to target tasks. To characterize this, we keep the learning method fixed and segment the one-dimensional task path space into $N$ tasks as shown in Fig. 4 and 5 and focus on how $N$ affects target task performance. We continuously transform tasks from source to target task, without violating the definition that all tasks lie on a task manifold that can be derived from the definition of a task[2]. All discretization experiments in this section were run with a maximum of 100 epochs and patience of 20 epochs across 3 random seeds. As visualized in Fig. 4, increasing discretization removes most

---

[1]Using 10% minimum contrast to ensure meaningful gradients.
[2]This rules out naive interpolation between two standard computer vision datasets.

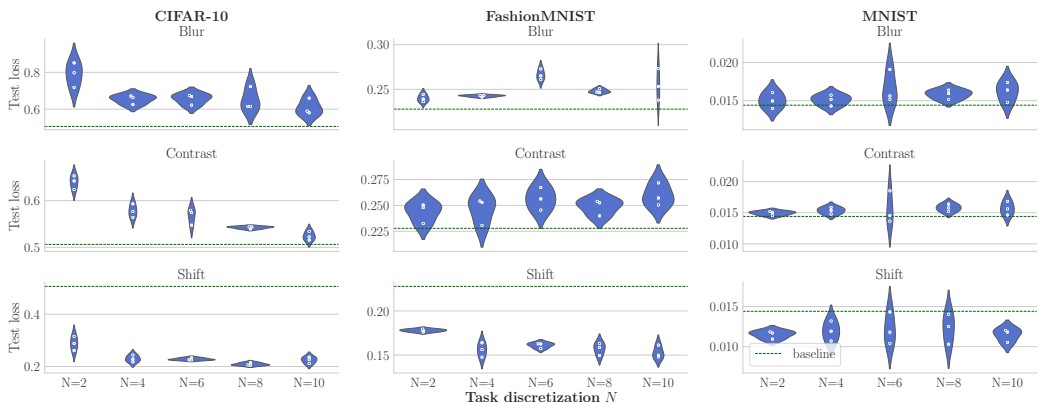

Figure 4: Final test loss on standard images as swarm and violin plots. The x-axis records the number of discretization steps, where $N = 2$ refers to training first, e.g. on blurred then on sharp images. Higher discretization provides a more gradual task shift from initial to final transformation. The y-axis shows the test loss on standard images with dots colored to distinguish different random seeds. The dotted line shows the mean baseline performance across 3 seeds. A clear decrease of negative pretraining is visible for CIFAR-10. On FashionMNIST, average results do not change noticeably except for *shift*. MNIST does not show clear negative pretraining.

of negative pretraining of blurring and contrast on CIFAR-10. Remarkably, the already improved results of random shift are further improved via discretization. On the other hand shows no clear trend on the FashionMNIST and SVHN datasets is visible. Possibly on these datasets, the "nudge" that increased discretization provides is not enough to overcome entrenched model characteristics.

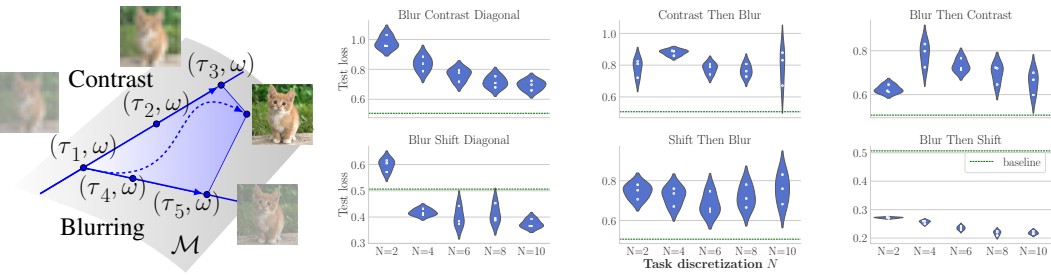

Figure 5: *Left:* Various two-dimensional sequential learning paths and discretizations from source to target task. *Right:* Results of deterministic 2D-paths with different discretizations on CIFAR-10. In most cases finer discretization improves results. We remark the surprising path-dependence of results.

**Multi-dim. paths from start to target task:** We extend the *sequential learning* setup to multiple dimensions as visualized in Fig. 5. Previously we only varied tasks along one transformation "dimension". Unlike the one-dimensional case, we now vary *both* contrast or random shift and blurring levels (see Fig. 5, left) at the same time. Each path starts with blurred images and either low contrast or high random shift transformation. Then we reduce transformations in the order mentioned in the title of each subplot.[3] We present results of these paths on CIFAR-10 in Fig. 5. Across most experiments and transformations, we see that performance on the target task increases as we approach a continuous density transformation between source and target tasks (i.e. as $N$ increases). We observe that continuous changes in tasks can help networks adapt to a new task ; in contrast, jumpy paths induce a distributional shift too large to overcome, leading to poor performance on the target task.

---

[3]*Example:* Blur then Contrast means that we start at blurred and low contrast images. The next tasks will first move to unblur the images, but keep them at low contrast. Once unblurred, task move to increase the contrast. *Diagonal:* Refers to lowering both transformations equally for each discretization step.

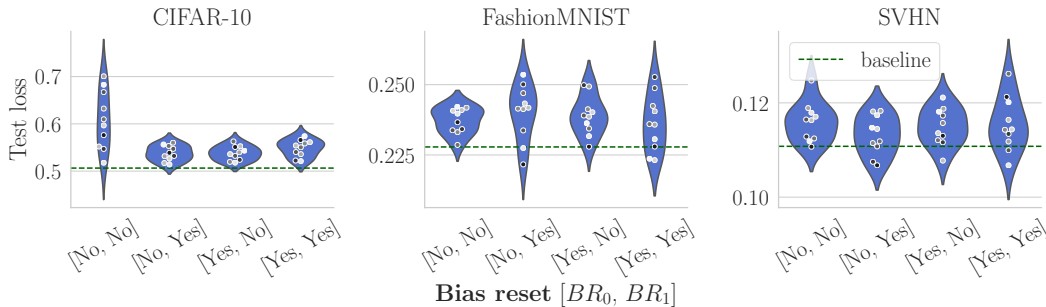

Figure 6: Final test loss on unblurred images as swarm and violin plot. The x-axis denotes whether bias resetting was applied before the first task with blurred images and/or the second task with unblurred images. The y-axis shows the test loss on unblurred images with dots colored to distinguish different random seeds. The dotted line shows the mean baseline performance across 3 seeds. A clear decrease of the negative pretraining effect is visible for CIFAR-10. On FashionMNIST and SVHN, overall a better performance is obtained however on average results do not change very much compared to the [No, No] negatively pretrained network.

## 6 BIAS RESETTING CAN PREVENT A LOSS OF PLASTICITY

The neural network literature is ripe with examples of initializing a model to be more amenable to optimization. For sequence modeling, the LSTM network is commonly initialized by setting biases such that the forget gate is inactive (Gers et al., 1999; Van Der Westhuizen & Lasenby, 2018). The same method has also found application in the first very deep networks such as Highway Networks (Srivastava et al., 2015) and its recurrent extension Recurrent Highway Networks (Zilly et al., 2017) to ensure that gradients flow through the entire network. Batch normalization (BN) (Ioffe & Szegedy, 2015) aims to normalize the bias and variance of network activations to improve training. Building upon this, Li et al. (2016) adapted to a new domain solely by retraining the BN weights while leaving all other parameters unchanged. Likewise we reason that resetting biases may, albeit shortly, activate many ReLU-activations in the network and thus allow gradients to flow freely.

In this section we focus on the effect of resetting the network biases. Again, we use a task path consisting of two tasks and blurring as transformation for all three datasets. Using *bias reset* (BR) implies, that before training on a task all biases are reset by adding $0.01$ to their value manually chosen to be of a similar order of magnitude as the largest biases in the network. Depending on which task *bias resetting* is used, we distinguish the following set-ups:

1. **No BR - No BR**: No *bias resetting* is used for both task, which is the default set-up.
2. **No BR - BR**: Only before training on the second task the network biases are reset.
3. **BR - No BR**: Just right in the beginning, before any training, the biases are reset.
4. **BR - BR**: The biases are reset before both training on the first task and the second task.

Again we used 10 different random seeds for the experiments and provide as baseline which is averaged across three random seeds. Viewing results in Fig. 6, we note that the removal of the negative pretraining effect is most pronounced on the CIFAR-10 dataset. Remarkable for this intervention is that resetting biases even before the blurred image task and not before training on sharp images ([Yes, No]) can remove the negative pretraining effect. On FashionMNIST and SVHN, the BR-intervention improves the test loss slightly ([Yes, Yes] vs. [No, No]). For these two datasets, the negative pretraining effect is however also less substantial. Notable however is that with BR intervention some seeds now are better than the baseline whereas before [No, No] this was not the case. One observed difference between CIFAR-10 and the two other datasets is the amount of training until best validation epoch. Bias reset leads to an increase in the number of epochs (about $+44\%$ for [Yes, Yes] compared to [No, No]) for CIFAR-10, while for FashionMNIST and SVHN the number of epochs decreases when using bias reset ($-44\%$).To corroborate our understanding of these results we provide norms of biases, weight as well activation statistics in appendix 8.3.

# 7 SPECULATION ON CAUSES OF THE NEGATIVE PRETRAINING EFFECT

What we have shown are three markedly distinct ways of removing negative pretraining effects. What remains unclear is why we observe such a learning and generalization behavior. In the following, we will attempt an explanation or speculation of the underlying causes of the observed learning phenomena. An important aspect to note before diving in deeper is that the negative pretraining effect is *not an optimization issue*. In all cases the training loss is reduced either to zero or to the loss it generally reaches upon convergence even when generalizing well. Negative pretraining causes a change in *generalization* on the test set but not a drop in performance on the training set.

**Large update rate phase:** A plausible explanation for the observed behavior is that the phase with larger updates similar to the *large learning rate phase* for neural networks is qualitatively different to updates in the low-loss setting. All interventions achieve settings where either A) a higher learning rate makes all updates larger, B) a higher loss setting is encountered more frequently, or C) resetting slows down convergence. Such a high learning rate phase is associated with better generalization.

More specifically, Zhu et al. (2018) claim that the anisotropic noise of SGD helps the network escape *sharp minima* and thus generalize better. The escape probability can be higher, if the training loss is still large compared to the loss at the minimum of the SGD optimization. Seong et al. (2018) evoke the same image, plausibly showing that large learning rates make it more likely to escape sharp minima. In the same spirit Nakkiran (2020) provides an example of how a mismatch between training and test loss landscape can favor optimization with an initially large learning rate. Lewkowycz et al. (2020) argue that in the large learning rate phase the curvature of the loss landscape is lowered thus yielding a tangible difference in optimization behavior. Focusing on bias resetting, centered biases are known to accelerate convergence (Clevert et al., 2015; Cun et al., 1991; LeCun et al., 2012; Schraudolph, 1998). Ensuring that biases are not centered is thus likely to have the opposite effect of slowing convergence which may ensure training remains in the large update rate phase longer.

**Preshaped learning landscape:** Taking the opposite viewpoint when gradient updates are not large, optimization is likely to occur within the "optimization valleys" in which the optimization started due to pretraining. In the limit case of a converged network, such a scenario is described for very wide neural networks (Lee et al., 2019). These wide networks are characterized by the *neural tangent kernel* (NTK). The largest eigenvalue of the NTK is in turn tied to generalization behavior (Lewkowycz et al., 2020; Jiang et al., 2019; Dyer & Gur-Ari, 2019). Possibly, the tangent kernel is not updated enough to overcome the *negative pretraining* effect when retraining on the target task. Corroborating this point, Oymak & Soltanolkotabi (2019) argue that SGD, under certain conditions, takes a direct path to the minimum such that final parameters may change little from initial parameters. Thus all interventions may serve to increase the training time, given the network more time and opportunity to adapt not only its output but also its tangent kernel to the target task.

# 8 DISCUSSION AND CONCLUSION

The proposed three interventions provide us with a host of insights on what determines plasticity as the ability to adapt in sequential learning. We learned that increasing the learning rate after pretraining can consistently benefit learning to the point of surpassing the performance of only training on standard data. In the discretization experiment setting, we learn that pretraining on another task can have a significant biasing effect on final results. By providing a finer discretization from source to target task, possible retraining issues are alleviated thus yielding a better target performance. Finally, perturbing the model after pretraining by resetting biases can similarly diminish the negative pretraining effect albeit not consistently. Resetting the biases of a network can increase the initial training loss for a new task and slow down convergence.

Several new questions arise from the experiments. What is the best task to start from? What is the best path of tasks and learning processes to follow thereafter? Which kind of model is best suited to make use of the advantages of a carefully-chosen path?

In this work, we presented a novel perspective on the sequential learning behavior of neural networks through the lens of negative pretraining. While casting light on issues a learning model may face in a sequential learning setup, we also discover a variety of new questions. The picture we are left with is an intriguing sequential learning problem setting offering a fruitful ground for further inquiry.

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
