# OpenReview forum: "The Negative Pretraining Effect in Sequential Deep Learning and Three Ways to Fix It"
_ICLR.cc/2021/Conference — Reject_

### Official Review · AnonReviewer2 · 2020-10-19
**Interesting hypotheses to mitigate negative pretraining effect**

**Rating:** 5
**Confidence:** 2

**Review:**

1. Summarize what the paper claims to contribute. Be positive and generous.
The paper claims to contribute three ways of mitigating negative pretraining:
  (1) altering the learning rate after pretraining,
  (2) increasing the discretization of data distribution changes from start to target task instead of "jumping" to a target task, and
  (3) resetting network biases to larger values.
Interesting hypotheses! Agreed that negative pretraining effect is a great topic to be studied further.

2. List strong and weak points of the paper. Be as comprehensive as possible.
 (1) Strengths
    a. The paper focuses on a single topic of exploring ways to mitigate negative pretraining effects.
    b. Easy to read and follow.
 (2) Weaknesses
    a. Empirical results aren't strong enough to back the three main claims; Need further analysis.
      i. For example, changing learning rate didn't seem to help with the learning task on FashionMNIST dataset.
      ii. Increasing the discretization of data distribution doesn't seem to improve the learning for (FashionMNIST, Contrast), (FashionMNIST, Blur), (MNIST, Blur), (MNIST, Contrast), and (MNIST, Shift).
      iii. The results of resetting network biases are not consistent across different datasets; need further analysis to make a more clear conclusion.
      iv. Some test loss differences seem too small and made me wonder about their statistical significance.

3. Clearly state your recommendation (accept or reject) with one or two key reasons for this choice.
Reject. Because of the aforementioned weaknesses, it is unclear what conclusion to make and apply to other research topics and/or applications.

4. Provide supporting arguments for your recommendation.
See 2.(2).a.

5. Ask questions you would like answered by the authors to help you clarify your understanding of the paper and provide the additional evidence you need to be confident in your assessment.
  (1) Could you provide further analyses to refute i-iv in 2.(2).a?

6. Provide additional feedback with the aim to improve the paper. Make it clear that these points are here to help, and not necessarily part of your decision assessment.
  (1) It will be helpful to qualitatively understand what the fine-tuned network learned if you visualize convolution features of both pretrained and fine-tuned models.

---

### Official Review · AnonReviewer4 · 2020-10-27
**An empirical investigation on the mitigation of “negative pretraining effect”; a good direction but not comprehensive enough**

**Rating:** 4
**Confidence:** 3

**Review:**

This work investigates the intriguing phenomenon where pretraining on one task hurts the finetuning performance of another. Besides being interesting in general, this phenomenon has practical relevance as pretraining becomes increasingly popular with large-scale models. Here, the authors present a clean case for “negative pretraining effect” on images, and propose three ways to mitigate it.

However, the investigation is limited to supervised image classification on four small scale datasets with blurring as the only deficit studied. The authors noted that the setup expanded that of Achille et al (2017); nonetheless, it would take more thorough experimentation to establish mitigations of a phenomenon, compared to merely observing one, especially given the empirical nature of this work. It’s not convincing to me that the proposed approaches are fundamental fixes that can be generally applied to other learning scenarios where one wishes to get rid of the “negative pretraining effect”.

The authors should consider expanding the set of “changes” they consider and experiment on a variety of tasks, ideally beyond vision, to establish the general efficacy of the approaches.

Some other comments: 1) The background on biological analogies is nice but has no explicit connection to the main idea of the paper; 2) Why is it interesting to apply “bias resetting” to the initial task, as the model starts from random anyway? 3) Why adding 0.01 to biases to “reset” it, which is hacky and hard to generalize, as opposed to “resetting” biases to 0? 3) The only case where “bias resetting” seems to help nontrivially is for CIFAR-10, but at which point this “resetting” is applied doesn’t seem to matter; why is that the case? Overall the results on “bias resetting” appear negative and perhaps shouldn’t be included as a mitigation, especially given that the idea itself lacks any concrete theoretical motivations.

---

### Official Review · AnonReviewer1 · 2020-10-29
**The paper is good written and easy to follow. The authors studied an important problem. More evaluation is desired for the studied interventions.**

**Rating:** 6
**Confidence:** 4

**Review:**

In this paper, the authors formalized the sequential learning problem and the negative transferring in this type of learning, and conducted empirical study on three interventions that can help to remove the negative transferring. Sequential learning (and its variants, such as continual learning and curriculum learning) is an important topic to study.

\+ formalization of sequential learning and the negative transferring of the learning, which facilitates future research on this topic

\+ Providing solid results (all on computer vision task though, more study on problems from diverse domains are desired) that demonstrates the high learning rate intervention is an effective approach to remove negative transferring.

\- The results for smoothing and bias resetting are somewhat limited. Out of the three datasets, only one clearly shows the impact of the intervention. To justify these two as useful approaches in general to removing negative pretraining effect, evaluation on more datasets and diverse learning problems is desired.

\- All studies were carried out on well controlled synthetic datasets. I understand that these datasets were generated from real life datasets and these well controlled studies are definitely needed.  However, a study (or more) of practical sequential learning problem is much needed to demonstrate the impact of the three interventions on real-word applications.

---

### Official Review · AnonReviewer3 · 2020-10-30
**The paper systematically investigates the sequential learning problem in neural networks through the lens of negative pretraining however there are several concerns relating to the experimental setup**

**Rating:** 4
**Confidence:** 4

**Review:**

**Summary:** This paper conducts an empirical study to examine the well-known negative transfer phenomenon (termed as a negative pretraining effect in this work) in neural networks. In particular, a network trained on a sequence of tasks performs inferior to a network trained from scratch on the intended target task. The main idea of the paper is to study this phenomenon by formulating and intervening on different constituents of the sequential learning process - (1) changing the learning rate across tasks, (2) number and type of tasks encountered in the learning process, and (3) resetting the model biases when going from one task to another. The paper conducts experiments on four visual classification datasets (CIFAR-10, FashionMNIST, MNIST, SVHN) and report their findings for sequential training of ResNet-18 architecture. They show that increasing the learning rate after training on the first task can alleviate the negative pretraining effect. They further showcase how different task discretization and resetting model biases help to reduce the effect.

**Pros:**
The paper investigates an important problem of negative pretraining which has implications for different time-dependent learning paradigms (lifelong/ continual learning). Systematically studying how different constituents of the sequential learning process affect the severity of the negative pretraining effect is a reasonable approach and this paper attempts to take an initial step with this approach.

**Cons:**
There are several concerns about the experimental setup which makes the results unconvincing:
- The paper examines a single model (ResNet-18) on four datasets and through different experiments, the paper demonstrates that on MNIST, FashionMNIST, and SVHN datasets there is no clear (or less substantial) negative pretraining effect. However, this paper is about how different interventions can help mitigate the negative pretraining effect. If the current experimentation setup does not render the phenomena (except for CIFAR-10), this raises the question of whether the paper is analyzing the right setup (datasets and model)? Effectively a single model (ResNet-18) is examined on the CIFAR-10 dataset for the interventions, which is not representative enough. What happens if we increase the model complexity? What happens if we change the input modality? (see [1] for more details)
- The paper concludes that increasing the learning rate on the subsequent task helps to **remove** the negative pretraining effect. In Figure 3 (discussion in Section 4) they report **a single case** of increasing the learning rate (10e-4 to 20e-4). It is unclear whether this is always the case. What happens if the learning rate is increased from 10e-4 to 50e-4 or 10e-4 to 10e-3? What is the general recipe to set the learning rate for the next task in the sequence?
- The paper claims that it has proposed three distinct ways to **remove** the negative pretraining effect. Given the empirical evidence, this **(remove)** is too strong a claim to make. In some experiments, it alleviates the negative effect (see Figure 3, 4, 6: CIFAR-10) while in other experiments, results are inconclusive (see Figure 5: CIFAR-10, Shift-Then-Blur, Contrast-Then-Blue, and most of the experiments on MNIST, FashionMNIST, SVHN).

**General comment:** There is no denying the fact that this paper studies an interesting phenomenon through systematic experiments. However, the authors should consider evaluating the proposed interventions on the setup (datasets/ models) where the negative pretraining is a clearly visible phenomenon to conclude generic applicability of the discussed interventions.

Please cite the below-mentioned work as it also empirically demonstrates the difficulty of warm-starting with pre-trained initialization.

[1] Ash, Jordan T., and Ryan P. Adams. "On the difficulty of warm-starting neural network training." arXiv preprint arXiv:1910.08475 (2019).

Please cite the peer-reviewed version of the related literature instead of the arXiv (for available ones), e.g.:

Dyer, Ethan, and Guy Gur-Ari. "Asymptotics of Wide Networks from Feynman Diagrams." International Conference on Learning Representations. 2019.

---

### Official Review · AnonReviewer5 · 2020-11-04
**A review**

**Rating:** 4
**Confidence:** 4

**Review:**

## 1. Brief summary
The authors study what they call a negative pretraining effect = models pretrained on task 1 and tuned on task 2 sometimes underperform compared to just training on task 2 from scratch. This is an important factor in many forms of life long learning, multi-task learning and curriculum learning. They investigate 3 potential remedies / setups: a) using different learning rates for task 1 and task 2, b) changing from task 1 to task 2 more smoothly, and c) resetting network biases at different stages of the process. The perform with a single ResNet18 architecture on MNIST, Fashion MNIST, SVHN and CIFAR-10.

## 2. Positive things
* I think the problem is very important and applicable in many problems in ML, especially in its practical deployment
* The problem is important and interesting from the theoretical point of view as well
* The interventions / experimental setup the authors use (a) LR changes, b) smooth task transition, and c) bias resetting) are all reasonable and potentially actionable if shown effective
* I like that the authors report results of multiple random seeds and show the full distribution of the results. This allows the reader to better judge the validity of the claims made and
* I like many aspects of your experimental setup
* I like the idea with path discretization and studying the effect of the number of discrete steps

## 3. Negative things / points of confusion

1. The experiments are way to limited in scope to establish generally applicable results. I'm not asking you to go all the way to ImageNet, but several things would greatly improve the potential value of your work:

a. Adding datasets with a different number of classes than 10. All the datasets you used have 10 classes (MNIST, Fashion MNIST, SVHN, CIFAR-10) and it is at least plausible that some of the effects you see could be caused by some sort of class specialization. It would be very helpful to see smaller number of classes (you can restrict some of the datasets you have already) and let's say CIFAR-100 on the other end. I think these would not be crazy hard to add but would greatly increase the value of your paper.

b. You only use a single architecture. It would be good to try others. Especially the learning rate considerations and experiments could be influenced by the fact that you chose a ResNet which are often hard to train from initialization at low learning rate (people sometimes preheat them by high LR initial phase of training). You could try a simple multi-layer CNN, and fully-connected multi-layer net. It would be good to see that your conclusions hold generally, not just for ResNets.

2. I am not extremely convinced by some features of your experiment A where you use different learning rates for the blurred task 1 and the sharp task 2. Firstly, why is the baseline performance a fixed LR = 10e-4? Did you use a hyperparameter search of sorts of find it? Because if it were optimal, the fact that e.g. panel 1 in Figure 3 the first two results perform worse could be just because their LRs are smaller, and not because of the effect you're trying to observe (the negative pretraining). It is at least suspicious that the blurred -- sharp task starts performing equally to the default sharp task *precisely* when the LR of the blurred -- sharp task is at least as big as the default task. It is certainly a confounding factor that does not allow you to draw the conclusion you do -- namely that it is the LR change that helps, rather than just increasing the LR. I suggest you show multiple baselines with different LRs so that we can compare to them. Also, please show the errorbars for the baseline performance as well, so that we can judge the overlap of the uncertainty intervals.

3. In Figure 3 there is essentially no signal there for Fashion MNIST and a week signal for SVHN. How does that fit with the causal explanation you are offering. I'm really not seeing how you can draw a conclusion as strong as "the precise order of learning rates across tasks can be crucial". I don't think Figure 3 shows that clearly, or possibly at all. Especially considering the potential confounding that I described in point 2.

4. For experiment A in Figure 3, wouldn't you expect that the ordering of the LRs would matter. In panel 1 [5,5] and [10,5] look the same, why? and [10,20] and [20,20] also look the same. How could both be true if you're expecting that you should either increase/decrease the LR to get good performance.

5. This is a point that I am not sure about, so please correct my if I'm wrong. I don't get how precisely you do the bias resetting. Do you set all biases to 0, or reinitialize them again from a distribution. Why would doing this at the begging before the blurred task have any effect. And if it does, doesn't it just mean that your original initialization was not good enough and now you made it better. It might not have much to do with the sequential learning but rather with this. Let me know if I'm wrong here.

6. In Section 7 you say "negative pretraining is not an optimization issue". That is possible, but I don't think you show that. The fact that it's a question of generalization and not the training loss going to zero does not mean it's not a question of optimization -- optimization includes the generalization properties of whatever optimum it found, so it definitely does deal with this. This might be an issue of semantics and if so just let me know, this is not a major point.

7. Figure 6 results don't look very conclusive to me, which probably means that the bias reset doesn't work too well (method C).

## 4. Tiny issues and suggestions (I'm not judging the paper based on those):
* You use e.g. 1e-1 instead of $1\times10^{-1}$ and it doesn't look very good (it's also kind of hard to read). It might be worth typesetting those values in a more aesthetic way.

## 5. Potentially relevant papers
When you discuss the effect of learning rate on updates, it sounds similar to this paper:
[1] Stiffness: A New Perspective on Generalization in Neural Networks (https://arxiv.org/abs/1901.09491) where they measured how "stiff" a NN is when trained with different LRs by comparing the learning effect of one image on another (also between datasts).

For the bias resets the class structure the DNN learns might be crucial. Here's a good overview of such class-specific effects:
[2] Traces of Class/Cross-Class Structure Pervade Deep Learning Spectra (https://arxiv.org/abs/2008.11865)

In the NTK discussion [3] discusses how the NTK behaves when you expand it around a partially pre-trained point in the weight space which sounds relevant to the way you study how the task 2 training depends on where task 1 got you:
[3] Deep learning versus kernel learning: an empirical study of loss landscape geometry and the time evolution of the Neural Tangent Kernel (https://arxiv.org/abs/2010.15110, NeurIPS 2020). It also discussed the importance of the early phases of training that you discuss.

[4] The Break-Even Point on Optimization Trajectories of Deep Neural Networks (https://arxiv.org/abs/2002.09572, ICLR 2020) also looks at the crucial effect of the early phases of training.

## 5. Conclusion
I think the question asked is interesting and the approach the authors took promising. However, the breath of experiments is not large enough and I think there are significant potential confounding effects in at least the setup A (changing the learning rate) that make it really hard / inconclusive to draw strong causal conclusions about the effect of the intervention on the negative pretraining effect. I encourage the authors to improve the paper and resubmit -- this seems like a really promising piece of research, but as it is it's not strong enough.

---

### Decision · Program_Chairs · 2021-01-07
**Final Decision**

**Decision:**

Reject

**Comment:**

Taking all reviews and the work in consideration, unfortunately the work does not present the breadth it needs to sustain the claims it makes. In particular, there work requires to analyse more architectures/variations of datasets with different properties and to provide more careful ablation studies that shows the efficiency of the 3 different proposed methods. Potentially removing one of this methods in order to give more space to analyse the others that seem more promising.